# Association of Low-Grade Glioma Diagnosis and Management Approach with Mental Health Disorders: A MarketScan Analysis 2005–2014

**DOI:** 10.3390/cancers14061376

**Published:** 2022-03-08

**Authors:** Debarati Bhanja, Djibril Ba, Kyle Tuohy, Hannah Wilding, Mara Trifoi, Varun Padmanaban, Guodong Liu, Michael Sughrue, Brad Zacharia, Douglas Leslie, Alireza Mansouri

**Affiliations:** 1Penn State College of Medicine, Hershey, PA 17033, USA; dbhanja@pennstatehealth.psu.edu (D.B.); ktuohy@pennstatehealth.psu.edu (K.T.); hwilding@pennstatehealth.psu.edu (H.W.); mtrifoi@pennstatehealth.psu.edu (M.T.); 2Department of Public Health Sciences, Penn State College of Medicine, Hershey, PA 17033, USA; djibrilba@phs.psu.edu (D.B.); gliu1@phs.psu.edu (G.L.); dleslie@phs.psu.edu (D.L.); 3Department of Neurosurgery, Penn State College of Medicine, Hershey, PA 17033, USA; vpadmanaban@pennstatehealth.psu.edu (V.P.); bzacharia@pennstatehealth.psu.edu (B.Z.); 4Centre for Minimally Invasive Neurosurgery, Prince of Wales Private Hospital, Randwick, NSW 2031, Australia; sughruevs@gmail.com; 5Penn State Cancer Institute, Hershey, PA 17033, USA

**Keywords:** glioma, low-grade, mental health disorder, quality of life, connectomics, resection, biopsy

## Abstract

**Simple Summary:**

Low-grade gliomas (LGGs) comprise 13–16% of glial tumors. As survival for LGG patients has been improving, it is important to consider the effects of diagnosis and treatment on mental health. The aims of this retrospective cohort study were to determine the incidence, prevalence, and risk factors of mental health disorders (MHD) in LGG patients. In our analysis including 20,432 LGG patients, we identified an MHD prevalence of 60.9%. Of those with no history of prior MHD, 16.9% of LGG patients developed a new onset of MHD within 12 months of LGG diagnosis. Risk factors included female gender, ages 35–54, presence of seizures, and first-line surgical treatment. Therefore, proactive surveillance and counseling surrounding MHDs are recommended among LGG patients. Impact of surgery on brain networks affecting mood should also be considered.

**Abstract:**

Low-grade gliomas (LGGs) comprise 13–16% of glial tumors. As survival for LGG patients has been gradually improving, it is essential that the effects of diagnosis and disease progression on mental health be considered. This retrospective cohort study queried the IBM Watson Health MarketScan^®^ Database to describe the incidence and prevalence of mental health disorders (MHDs) among LGG patients and identify associated risk factors. Among the 20,432 LGG patients identified, 12,436 (60.9%) had at least one MHD. Of those who never had a prior MHD, as documented in the claims record, 1915 (16.7%) had their first, newly diagnosed MHD within 12 months after LGG diagnosis. Patients who were female (odds ratio (OR), 1.14, 95% confidence intervals (CI), 1.03–1.26), aged 35–44 (OR, 1.20, 95% CI, 1.03–1.39), and experienced glioma-related seizures (OR, 2.19, 95% CI, 1.95–2.47) were significantly associated with MHD incidence. Patients who underwent resection (OR, 2.58, 95% CI, 2.19–3.04) or biopsy (OR, 2.17, 95% CI, 1.68–2.79) were also more likely to develop a MHD compared to patients who did not undergo a first-line surgical treatment. These data support the need for active surveillance, proactive counseling, and management of MHDs in patients with LGG. Impact of surgery on brain networks affecting mood should also be considered.

## 1. Introduction

Low-grade gliomas (LGGs, WHO Grade II gliomas) comprise 13–16% of all glial tumors [1]. The most common age of presentation is in the fourth and fifth decades of life, affecting individuals during some of their most productive years [1]. LGGs comprise a heterogeneous group of tumors that inevitably undergo malignant transformation, leading to neurological deterioration and death [2]. Consequently, the mean length of survival of patients with LGGs has classically been less than 10 years post-diagnosis, ranging between 3.2 and 7.7 years among different prognostic groups [3].

Due to their frequent origin in eloquent brain areas and relatively mild or absent neurological symptoms early on, LGGs have historically been treated with a wait-and-see approach to mitigate risk [3]. However, mounting evidence suggests a more aggressive treatment regimen, namely through upfront maximal safe resection and adjuvant therapy, which delays malignant progression and prolongs overall survival (OS) [4,5,6,7]. As the profile of LGG shifts to a more chronic condition with gradually improving survival, it is essential that the effects of diagnosis and treatment on quality of life (QoL) be considered.

One important aspect of quality of life that is largely studied among cancer outcomes is mental health [8]. Previous studies have concluded that there are significant associations between mental health disorders (MHDs) and the etiologies and prognoses of various subtypes of cancer [9]. The prevalence of depression and anxiety is also high among cancer patients compared with the general population (20% vs. 5% for depression and 10% vs. 7% for anxiety) [9,10]. Specific to brain tumor patients, the prevalence of depression is 21.7%, with patients described as having significantly greater distress and lower positive affect compared to those with other cancer diagnoses [11,12,13]. For cancer patients specifically, often preceding an MHD diagnosis is cancer-related distress, which is a multifactorial unpleasant experience affecting nearly one half of all cancer patients; it serves as an important risk factor for developing MHD [14]. Unmanaged MHDs can be detrimental to QoL, negatively modulate cancer treatment response, and serve as an important risk factor for low overall survival among patients with cancers [15].

Literature investigating the prevalence of MHDs among glioma patients remains limited to high-grade gliomas and their association with only depression or anxiety [16,17,18]. Furthermore, very few studies have assessed the effects of surgical management on MHD onset, and the studies available have demonstrated conflicting results [17,19,20,21]. The objective of this study was to leverage a commercial healthcare claims database to: (1) describe the prevalence of a broad spectrum of MHDs among LGG patients, (2) determine if there is a temporal correlation between LGG diagnosis and increased MHD incidence, and (3) identify associated risk factors for MHDs among the LGG cohort, including first-line treatment modality and presence of seizures.

## 2. Materials and Methods

The IBM Watson Health MarketScan^®^ Commercial Claims and Encounters database was queried to collect data from patients who were diagnosed with LGG from 1 January 2005 to 31 December 2014. This database contains all claims, both paid and adjudicated, at an individual level for those enrolled in employer-sponsored health plans in the United States (US) [22]. The end date of the study period was selected based on the transition from the International Classification of Diseases, Ninth Revision, Clinical Modification (ICD-9) to the 10th revision, which occurred in 2015. This avoided potentially confounding our data by including two different iterations of the classification system. MarketScan has been shown to be generally representative of commercially insured patients when compared with the Medical Expenditure Panel Survey [23].

Patients over the age of 18 diagnosed with LGG were included in this analysis. Patients were only included if they had at least 12 months of pre-LGG diagnosis enrollment and at least 12 months of post-LGG diagnosis enrollment to allow for adequate time to measure new onset of MHDs in the post-diagnosis period, as described in this study. We used ICD-9 codes to identify patients with LGG and MHDs. A diagnosis of LGG was determined using the ICD-9 code of 225.0. Patients were excluded if they had a code for brain metastasis (198.3) or meningioma (225.2) prior to their LGG diagnosis. The MHDs analyzed in this study included episodic mood disorders, including depression (ICD-9-CM codes 296.0–296.9), anxiety (ICD-9-CM codes 300.0–300.9), non-alcohol drug dependence (ICD-9-CM codes 304.0–304.9), adjustment reaction (ICD-9-CM codes 309.0–309.1), and depressive disorder not otherwise specified (ICD-9-CM codes 311.0–311.9). To capture those without an official MHD diagnosis, medication dispensing records were used for the psychotropic medications, including antidepressants, antipsychotics, and monoamine oxidase inhibitors, listed in Appendix A.

Prevalence of MHDs among LGG patients was measured by quantifying the number of patients with the presence of both an LGG code and MHD codes(s)/psychotropic drug fills at any point in their claims record. New onsets of MHD associated with LGG diagnosis were measured by quantifying the number of patients who had their first recorded MHD ICD-9 code or psychotropic drug fill after LGG diagnosis. We considered an MHD present after LGG diagnosis if found in the claims records within 12 months after the index diagnosis; this timing captured diagnoses likely associated with the cancer diagnosis and treatment, while minimizing other confounding life events. Covariates of interest included age (18–34, 35–44, 45–54, or 55–64 years), sex, US geographic region (northeast, north central, south, or west), presence of glioma-related seizures (identified by anti-epileptic drug fills listed in Appendix A), and first-line surgical treatment (identified by the presence of certain CPT codes—resection: 61510, 61518, 61305, 61304; biopsy: 61750, 61751, 61781—listed within 12 months of LGG diagnosis). “No surgical treatment” was defined as the absence of biopsy and resection codes within 12 months from LGG diagnosis. Seizure medications that were indicated for both MHD and seizures, such as clobazam, were excluded from the query.

Raw data are presented using descriptive statistics. Continuous variables are presented using mean and standard deviation. Dichotomous data are presented as frequencies and percentages. Comparison of continuous data was conducted using pooled, unpaired *t*-statistic testing and categorical data comparison with chi-square analysis or Fisher exact tests where appropriate. The level of accepting statistical significance was set at 0.05. The sample dataset was constructed using SPSS. Unadjusted odds ratio (OR) estimates and Wald 95% confidence intervals (CIs) were used to calibrate the strength of association between risk factors using logistic regression.

## 3. Results

### 3.1. Prevalence of MHD in LGG Patients

A total of 20,432 patients diagnosed with LGG from 2005 to 2014 were included in this study. Cohort sociodemographic information is summarized in Table 1. Of these patients, 12,436 (60.9%) were diagnosed with or were receiving treatment for at least one mental health disorder, either before or after LGG diagnosis.

### 3.2. New Onset of MHDs Post-LGG Diagnosis

To evaluate for new onset of MHDs post-LGG diagnosis, patients with no record of MHD pre-LGG diagnosis were queried. In total, 11,458 LGG patients were identified as never having a mental health disorder prior to LGG diagnosis. Of these patients, 1195 (16.7%) developed their first MHD within 12 months post-LGG diagnosis (Table 2). MHD incidence was significantly associated with female gender (OR, 1.14, 95%, 1.03–1.26), the western U.S. geographic region compared to the south (OR, 1.23, 95%, 1.06–1.42), and age range of 35–44 (OR, 1.20, 95%, 1.03–1.39) compared to 18–34 (Figure 1).

### 3.3. Impact of Glioma-Related Seizures on MHD Prevalence

For patients with no history of MHD prior to LGG diagnosis, the association of glioma-related seizures with new onset of MHDs was assessed using unadjusted ORs. In total, 11,458 LGG patients were included in this analysis, where 1799 (15.7%) experienced glioma-related seizures within 12 months of LGG diagnosis. A total of 494 (27.5%) of the patients with seizures developed an MHD. The “no seizure” group served as the reference point. Patients who experienced seizures were 2.1 times more likely to develop an MHD compared to patients who did not experience seizures (OR, 2.19, 95% CI, 1.95–2.47), summarized in Figure 1.

### 3.4. Impact of First-Line Treatment on MHD Prevalence

For patients with no history of MHD prior to LGG diagnosis, the association of first-line treatment modalities with new onset of MHDs was assessed (Figure 2). First-line treatment modalities were stratified by biopsy, up-front resection, or no surgical treatment. A total of 11,458 LGG patients were included in this analysis, where 313 (2.7%) underwent biopsy, 750 (6.6%) underwent resection, and 10,395 (90.7%) underwent no surgical treatment. The non-surgical treatment modality served as the reference point for comparison with all other treatments. Of patients who underwent resection, 236 (31.5%) developed MHD. Additionally, 87 (27.8%) patients who underwent biopsy and 1569 (15.1%) patients with no surgical treatment developed MHDs. Summarized in Figure 2, patients who underwent resection were more likely to develop an MHD compared to patients who did not undergo a surgical treatment (OR, 2.58, 95% CI, 2.19–3.04). Likewise, patients who underwent biopsy were more likely to develop an MHD compared to patients in the non-surgical treatment group (OR, 2.17, 95% CI, 1.68–2.79).

## 4. Discussion

In this large retrospective cohort study, we found that the prevalence of MHDs among LGG patients was 60.9%, which stands well above the national average (21.0%) [24]. Additionally, we found that 16.9% of patients with no history of an MHD went on to develop at least one in the year following LGG diagnosis. Independent risk factors included female gender and patients aged 35–44 at the time of LGG diagnosis. First-line treatment modalities involving surgical intervention, such as stereotactic biopsy and up-front resection, showed a greater association with an MHD diagnosis compared to initial non-surgical interventions. Finally, presence of glioma-related seizures was associated with an MHD diagnosis. The MHD prevalence estimated here for LGG stands well above the corresponding prevalence reported for cancer patients overall [9].

Drawing direct comparisons between our results and other glioma-specific studies is difficult because most studies focused on the prevalence of only depression or anxiety in glioma [16,17,18], whereas our study analyzed a broad spectrum of mental health disorders, including depression, anxiety, non-alcohol substance dependence, adjustment reaction, and non-specific depressive symptoms. Nonetheless, our estimate of the MHD prevalence in LGG patients (60.9%) is higher compared to other studies reporting depression rates in glioma patients. A review by Rooney et al. of 4089 glioma patients found between 6 and 28% of patients screen positive for depression [16]. Similarly, a review by Huang et al. found 21.7% of brain tumor patients had depression or depressive symptoms [13]. Additional studies among the brain tumor population have placed rates of anxiety between 30% and 60% [13,21].

Several studies have focused their analyses on high-grade glioma or collated the results of various intracranial tumors [21,25,26]. It is imperative that mental health outcomes of LGG are investigated discretely from high-grade gliomas or other brain tumors because LGGs represent a distinct pathological entity with varying growth patterns and clinical features [27]. For example, gliomas have unique relationships with brain networks; they integrate within synapses and likely cause more changes to neuronal connections more than other intracranial tumors do [28]. Furthermore, while LGGs eventually undergo malignant transformation, the survival for these patients is notably increased compared to high-grade gliomas. Patients may live for a significant period, during which LGG causes impairments to their mental health and QOL even before malignant transformation, either through disease progression or treatment [29]. As the LGG treatment paradigm shifts from conservative to more aggressive approaches to promote overall survival, the neuro-oncology community places an increasing emphasis on QoL and mental health of these patients. Our study represents the largest low-grade glioma patient cohort evaluated for MHDs reported in the literature [16]. This large-scale analysis is likely more representative of the true MHD prevalence estimate, with increasing geographic, institutional, and patient diversity.

There are inconsistent findings reported in the literature surrounding the impact of surgical treatment on MHD onset in patients with gliomas. Few studies have evaluated the effects of surgery extent on depression onset. Pringle et al. and Pelletier et al. demonstrated no association between extent of surgery and depression [19,20], while Mainio et al. reported lower rates of depression among patients who underwent gross total tumor resection [21]. Litofsky et al. reported lower rates of depression among patients with gross total resection compared to stereotactic biopsy based on findings from the “Glioma Outcomes Project” [17]. Our study, contrarily, showed a higher association of surgical interventions with MHD onset compared to non-surgical approaches. Again, direct comparisons are difficult, given these previous studies combined results from various brain tumors, focused only on depression and/or anxiety, and suffered from low patient number cohorts.

While surgery may not have a direct causative association with MHD prevalence, this relationship needs to be further explored. Prior literature postulates that craniotomy may incite or possibly alleviate significant impact on neuronal networks. Litofsky et al. proposed that surgery may decompress tumors, disrupting or compromising limbic pathways, and ultimately decrease the likelihood of patients developing MHDs [30]. Conversely, these limbic pathways surrounding tumor tissue may sustain damage or disruption following surgery, resulting in pathway impairment and consequent MHD onset. Similarly, resting state functional connectivity may be impaired after surgical resection. Sparacia et al. demonstrated that intracranial surgical resections, despite no obvious complications or clinical deterioration in patients, resulted in decreased functional brain connectivity from pre-surgical resection to post-surgical resection when measured with resting functional magnetic resonance imaging (fMRI) [31]. Interestingly, scientists have corroborated changes in resting-state network connectivity and symptom-linked pathophysiology with four different biotypes of depression [32]. Though no relationship between surgical resection, resting-state network changes, and depression has been proven in the literature, it is plausible that changes in functional connectomes as a result of intracranial resection may influence the onset of depression in brain tumor patients. Herein lies the value of connectome atlases to disentangle this relationship and, perhaps, utilize these detailed maps for surgical planning and patient counseling [33].

Furthermore, upfront resection may be correlated with more advanced-stage tumors. Reactions to advanced cancer diagnoses, in general, may consist of shock, disbelief, despair, anger, etc., which could manifest into mental health disorders, such as adjustment reaction or anxiety [25,34,35]. Prolonged adjustment reactions can develop into major depressive episodes. Finally, patients undergoing major surgical procedures experience increased fear and anxiety, which is even more evident during awake craniotomy procedures [36]. Nonetheless, given the conflicting nature of these results, prospective longitudinal studies, with emphasis on tumor location, recurrence, and specific MHD diagnosis, are necessary to augment our understanding of the relationship between surgical treatment of LGGs and MHD onset.

Broadly, lifestyle detriments associated with LGG disease progression may have profound impacts on MHD onset and QoL [8,37]. Given the frequent location of LGG tumors in eloquent brain areas, patients may suffer from dysfunction in speech, motor skills, ambulation, vision, and cognition, among many other things, and these deficits can present even during stable disease periods [37]. Patients may have decreasing levels of social and occupational functioning and become withdrawn and reduce interaction with others [30]. In addition, as demonstrated in our findings, patients who suffered from glioma-related seizures were more likely to develop MHDs compared to patients who did not experience seizures. Frequent seizures have been associated with anxiety disorders and are presumed to negatively impact QoL [38]. All these psychosocial and neurological aspects are theorized to contribute to the multifaceted mental health disorder onset seen among LGG patients.

Findings from this analysis support the need for increased awareness, active surveillance, and proactive psychosocial counseling and cognitive management of MHDs in LGG patients. Mental health issues are known to be undermined in the cancer care domain, as clinicians may misidentify symptoms (hopelessness, helplessness, excessive worrying) as normal emotional responses to a cancer diagnosis, consequently leaving patients with inadequate mental health provisions [39]. Not only do cancer patients with co-morbid MHDs have poorer health outcomes but they also have decreased adherence to treatment. Treatment adherence is especially important for LGG patients, who often have weekly cycles of radiation therapy or chemotherapy, which require regular outpatient and follow-up visits. Furthermore, recent studies have demonstrated that more than half of glioma patients report psychological distress during their disease course [40], and a majority of brain tumor patients medically require psycho-oncologic services. While underutilized in comprehensive cancer care, psychosocial support services markedly move the needle in decreasing depressive symptoms, improving self-reported levels of existential and functional well-being, and improving global quality of life, as shown in a recent randomized controlled trial involving 50 brain tumor patients [41]. Many neurosurgeons and neuro-oncologists do not incorporate MHD evaluation or diagnosis in their management of LGG patients. We implore these providers to actively incorporate this into their routine assessments. MHD screening and counseling among LGG patients starting immediately post-diagnosis should be implemented as best practices in providing holistic care to these patients.

There are limitations to this study that are inherent to any study using a commercial claims database, which is designed for billing purposes, not for clinical use. Our study relied on the accuracy of each healthcare organization’s ICD-9 and CPT coding, which provides a potential source of inaccurate or incomplete reporting. This study is one of the first to evaluate a LGG patient population using the MarketScan research database, so these results should be interpreted under this context. It is also one of the first studies to measure MHDs among LGG patients using commercial claims data in this indirect approach. However, published studies have used similar methodologies to explore MHD in patients with benign meningioma, non-functional pituitary adenomas and head and neck cancers, demonstrating a validated method [42,43,44]. Additionally, in our cohort queries, we did not exclude patients with comorbid or prior cancer history, to be representative of the true LGG population. However, this may influence the MHD prevalence rates, as increased tumor burden may increase likelihood of MHD. Regarding ICD-9 codes, prior to 2011, non-specific biopsy codes were utilized for all intra-axial, extra-axial, and spine procedures (61750, 61751, 61781). While these were not specific to stereotactic biopsy, they were queried within 12 months of LGG diagnosis, which increased the likelihood that the biopsy was related to LGG. Finally, the 2005–2014 timeframe of this study is a limitation. Much has been learned in the last decade about LGG treatment and management; however, as with many other guidelines, there is a likely gap before LGG-specific recommendations, including maximal safe resection upfront, are incorporated into routine clinical practice [45,46,47]. We hypothesize that our estimates of MHD prevalence and risk factors here are similar to those of the present day.

MHD prevalence may have been underestimated in this analysis. Mental health services are reported to be underutilized by patients with psychiatric symptoms. In this study, only clinician-reported MHDs were recognized through ICD-9 diagnosis codes and psychotropic drug fills. Similar studies investigating MHD prevalence among glioma patients have used patient self-assessments of depression as markers for MHD [17]; however, this was not possible in the current study design. Additionally, patients who experienced cancer-related distress were not identified in this analysis. Given the limitations of our database, we were not able to query for subclinical manifestations of poor mental health that are not represented by ICD-9 or CPT codes. Cancer-related distress and their association with subsequent MHD onset should be measured in future studies. Moreover, we did not evaluate the presence of bipolar disorder, primary psychosis, or various other psychiatric diagnoses because these have not been associated with other cancers; excluding them here, in addition to the lack of Medicare and Medicaid patient representation in the commercial claims database, could have led to an underestimation of MHD prevalence in LGG patients.

## 5. Conclusions

Prevalence of MHD among LGG patients was estimated to be a tremendous 60.9% in this large population-based retrospective cohort study. We found that 16.9% of patients with no history of MHD developed one within the post-LGG diagnosis period, most significantly affecting females, those aged 35–44 at the time of LGG diagnosis, those with glioma-related seizures, and those undergoing first-line surgical treatment of their tumors. Given the high prevalence of MHDs, suggested best practices include active surveillance and psychosocial counseling following LGG diagnosis in order to provide holistic care. Future work should incorporate connectome atlases in surgical planning and patient counseling.

## Figures and Tables

**Figure 1 cancers-14-01376-f001:**
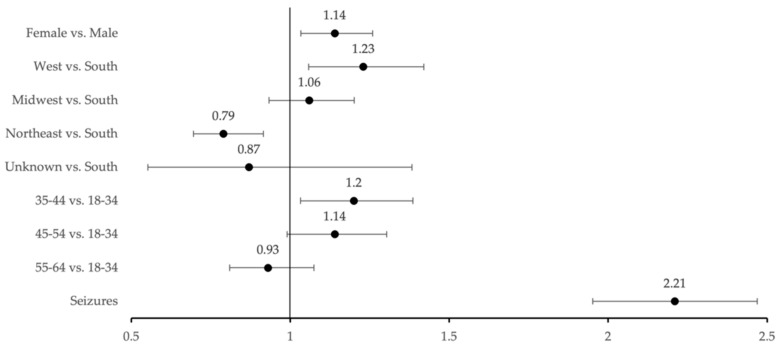
Risk factors associated with increased MHD incidence post-LGG diagnosis using unadjusted odds ratios with 95% Wald confidence intervals.

**Figure 2 cancers-14-01376-f002:**
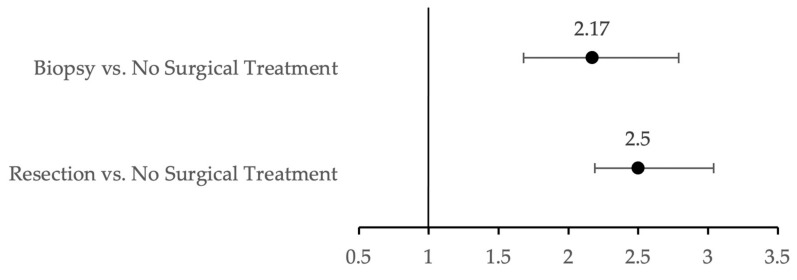
First-line treatment associated with increased MHD incidence post-LGG diagnosis using unadjusted odds ratios with 95% Wald confidence intervals.

**Table 1 cancers-14-01376-t001:** Baseline characteristics of patients with low-grade glioma.

Characteristic	No. (%) of Patients
**Age, years**	
18–34	4297 (21.03)
35–44	4191 (20.51)
45–54	5995 (29.34)
55–64	5949 (29.12)
**Sex**	
Male	8105 (39.67)
Female	12,327 (60.33)
**US Region**	
South	8392 (41.07)
West	2763 (13.52)
Midwest	5020 (24.57)
Northeast	4019 (19.67)
Unknown	238 (1.16)
**Mental Health Disorder**	
Yes	12,436 (60.9)
No	7996 (39.1)

**Table 2 cancers-14-01376-t002:** Sociodemographic and clinical comparison of low-grade glioma patients with and without mental health disorder onset within 12 months post-LGG diagnosis.

	Study Group, No. (%) of Patients	
	No MHD Post-LGG Diagnosis	MHD Post-LGG Diagnosis
**Characteristic**	**9543 (83.3)**	**1915 (16.7)**
**Age, years**		
18–34	2241 (23.48)	425 (22.19)
35–44	1918 (20.10)	435 (22.72)
45–54	2720 (28.50)	574 (29.97) *
55–64	2720 (28.50)	481 (25.12)
**Sex**		
Male	4476 (46.90)	836 (43.66)
Female	5067 (53.10)	1079 (56.32) *
**US Region**		
South	3926 (41.14)	791 (41.31) *
West	1223 (12.82)	302 (15.77)
Midwest	2166 (22.70)	462 (24.13)
Northeast	2103 (22.04)	338 (17.65)
Unknown	125 (1.31)	22 (1.15)

* Denotes significant findings, where *p* < 0.05.

## Data Availability

Data supporting reported results is available on request from the corresponding author.

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
