# Peer review of "Association of Low-Grade Glioma Diagnosis and Management Approach with Mental Health Disorders: A MarketScan Analysis 2005–2014"

_cancers, 2022, doi:10.3390/cancers14061376_

Round 1

Reviewer 1 Report

Thank you for the opportunity to read “Association of Low-Grade Glioma Diagnosis and Management Approach with Mental Health Disorders: A MarketScan Analysis 2205-2014.” This manuscript provides valuable information regarding a vulnerable population. However, I am questioning the reporting on results 8+ years prior. Much has been learned about the LGG population from 2014 -present, including their treatment (e.g., commonality of surgical resection) and mental health / management options. I understand the rationale for the 2015 cut date – but a better design would be to complete both (<2015 / >2015) and then compare to see If prevalence / risk factors hold true. As such, some revisions are needed prior to acceptance.

Introduction 

-Mental health in oncological populations extends beyond psychological health (e.g., depression and anxiety) – in fact, more prevalent in this population is cancer-related distress (e.g., death-distress, fear of tumor recurrence, and death anxiety). Literature has recently uncovered the prevalence and risk factors of such distress – with a goal of intervention creation. This is a significant void from the manuscript. While I understand there are no specific ICD-9 diagnosis codes for cancer-related distress, this needs to be classified in the introduction that the focus was only on psychological distress – which is less prevalent than cancer-related distress in glioma patients.

Methods

-Who were the authors of these clinical notes – we know many physicians (neurosurgeon, neuroradiologist, neuro-oncologist) do not inquire about MHD let alone diagnosis MHD. This is a significant limitation of this study and needs to be outwardly identified, not just tucked into the limitations section.

Results

-Why only examine MHD 12 months post diagnosis for new onset? In some of these patients you have data that extends well beyond 12 months. I recommend you look longitudinally to see if there is any association with time since diagnosis and MHD diagnosis. Associated – can you look at tumor recurrence and its impact on MHD?

-The finding of seizures associated with MHD is remarkable!

-I am slightly shocked, even in 2005-2014 that the incidence of no surgical intervention was 10,395 (90.7%). How was the diagnosis of LGG given without even a biopsy?

Conclusions

-Fabulous job tying this data together, grounding it in current literature, and identifying the limitations which caution the overinterpretation of results.

Reviewer 2 Report

This is a retrospective cohort study of >20,000 low grade glioma patients, which aimed to assess the incidence, prevalence, and risk factors of mental health problems. The manuscript is well written, and the analysis is appropriate for the research questions, which adds significant new knowledge to a growing field of concern for historically underappreciated health problems in low grade glioma patients. I have few queries:

  1. Could there be individuals in the database who did not declare their history of LGG cancer. Were patients with other cancers identified and/or excluded?
  2. Could perhaps briefly explain MEPS for anyone outside of the US (E.g. perhaps more simply national census data?)
  3. If data are available it would be valuable to understand the nature of MHDs by each ICD-9-M category if possible (and not just collectively, as the presence of any MHD diagnosis) to better understand the specific challenges, which would help to facilitate comparisons with other studies conducted and, on beyond this, help ascertain if these were similar/different to newly diagnosed MHD post-diagnosis, or if they remained similar over time during the period studied.
  4. It would be interesting also to look at more long-term data (>12 months) to understand the lasting effect on patient’s mental health. 

Round 2

Reviewer 1 Report

All recommendations have been appropriately addressed. Job well done. !